# Harmonization of Newborn Screening Results for Pompe Disease and Mucopolysaccharidosis Type I

**DOI:** 10.3390/ijns9010011

**Published:** 2023-02-27

**Authors:** M. Christine Dorley, George J. Dizikes, Charles Austin Pickens, Carla Cuthbert, Khaja Basheeruddin, Fizza Gulamali-Majid, Paul Hetterich, Amy Hietala, Ashley Kelsey, Tracy Klug, Barbara Lesko, Michelle Mills, Shawn Moloney, Partha Neogi, Joseph Orsini, Douglas Singer, Konstantinos Petritis

**Affiliations:** 1Tennessee Department of Health, Division of Laboratory Services, Nashville, TN 37243, USA; 2College of Health Sciences & Public Policy, Walden University, Minneapolis, MN 55401, USA; 3Tennessee Department of Health, Division of Laboratory Services, Knoxville, TN 37920, USA; 4Division of Laboratory Sciences, National Center for Environmental Health, Centers for Disease Control and Prevention, Atlanta, GA 30341, USA; 5Illinois Department of Public Health, Chicago, IL 60612, USA; 6Maryland Department of Health, Baltimore, MD 21205, USA; 7Virginia Department of General Services, Division of Consolidated Laboratory Services, Richmond, VA 23219, USA; 8Minnesota Department of Health, St. Paul, MN 55155, USA; 9Michigan Department of Health & Human Services, Lansing, MI 48906, USA; 10Missouri State Public Health Laboratory, Jefferson City, MO 65101, USA; 11Department of Pathology, Indiana University, Indianapolis, IN 46202, USA; 12Kansas Health and Environmental Laboratories, Topeka, KS 66620, USA; 13California Department of Public Health, Richmond, CA 94804, USA; 14Wadsworth Center, New York State Department of Health, Albany, NY 12208, USA; 15Ohio Department of Health, Reynoldsburg, OH 43068, USA

**Keywords:** harmonization, pompe, MPS I, newborn screening, digital microfluidics, tandem mass spectrometry, multiples of the median, regression

## Abstract

In newborn screening, false-negative results can be disastrous, leading to disability and death, while false-positive results contribute to parental anxiety and unnecessary follow-ups. Cutoffs are set conservatively to prevent missed cases for Pompe and MPS I, resulting in increased falsepositive results and lower positive predictive values. Harmonization has been proposed as a way to minimize false-negative and false-positive results and correct for method differences, so we harmonized enzyme activities for Pompe and MPS I across laboratories and testing methods (Tandem Mass Spectrometry (MS/MS) or Digital Microfluidics (DMF)). Participating states analyzed proofof- concept calibrators, blanks, and contrived specimens and reported enzyme activities, cutoffs, and other testing parameters to Tennessee. Regression and multiples of the median were used to harmonize the data. We observed varied cutoffs and results. Six of seven MS/MS labs reported enzyme activities for one specimen for MPS I marginally above their respective cutoffs with results classified as negative, whereas all DMF labs reported this specimen’s enzyme activity below their respective cutoffs with results classified as positive. Reasonable agreement in enzyme activities and cutoffs was achieved with harmonization; however, harmonization does not change how a value would be reported as this is dependent on the placement of cutoffs.

## 1. Introduction

Emphasis on quality improvement (QI) processes has increased in recent years for newborn screening (NBS) laboratories. The US federal government has recommended turnaround times for reporting screening results to improve time to intervention for disorders detectable in the newborn period [1,2]. Coupled with these QI processes has been increased scrutiny regarding laboratory cutoffs that contribute to false-negative results and missed cases leading to disability and even death [1,3,4]. Disorders recently added to the Recommended Uniform Screening Panel (RUSP) [5] all have phenotypic spectrums including X-linked adrenal leukodystrophy (X-ALD), Pompe disease (PD), and mucopolysaccharidosis type I (MPS I), which may contribute to increased false-positive screening reports [6,7,8] and potential for unnecessary follow-ups and diagnostic testing.

NBS for PD presents challenges related to the detection of late-onset variants in patients who may never develop disease [9,10,11]. This situation creates significant ambiguity and anxiety for families [12,13]. Late-onset or attenuated forms of MPS I and variants of unknown significance with undetermined pathogenicity also create parental stress and psychosocial burden [7,14]. Benign pseudodeficiencies caused by low enzyme activities (e.g., α-glucosidase (GAA) for PD and α-L-iduronidase (IDUA) for MPS I) and the presence of carriers pose challenges for laboratories screening for these disorders [15,16], since these low activities contribute significantly to high false-positive rates and lower positive predictive values (PPV) [7,8]. While the RUSP serves as a guide for establishing state NBS panels [5], NBS programs ultimately decide which diseases to screen for and place their screening cutoffs conservatively to avoid missed cases [10,12]. Consequently, some programs report more false-positive results than others [10,17]. False-positive reports, which also contribute to parental anxiety [11,12,14,18] and over utilization of health care systems [1], can be reduced by way of second-tier assays, such as measuring the level of glycosaminoglycans (GAGs) for MPS I [19,20,21], evaluating the ratio of GAA activity to creatine/creatinine for PD [16,22], and using post-analytical tools such as Collaborative Laboratory Integrated Reports (CLIR) developed by the Mayo Clinic [15,20,23].

Data harmonization is also suggested to reduce false-negative and false-positive results, correct for method differences, and achieve greater uniformity of results and clinical conclusions [3,4,6,24,25]. Harmonization, considered to be a “holy grail” [25], facilitates scaled comparisons and is “the process of recognizing, understanding, and explaining differences while taking steps to achieve uniformity of results, or at minimum, a means of conversion of results such that different groups can use the data obtained from assays interchangeably” [26]. Some NBS labs have successfully used multiples of the median (MoM) to harmonize results for MPS I [27,28], TREC [29], and in a large study involving homocystinuria results from 32 NBS programs covering 18 countries [17]. Other methods have also been employed, such as using quality control materials and regression equations to harmonize NBS proficiency testing (PT) results for amino acids and acylcarnitines [24]. Barriers to harmonization efforts have been attributed to differences in instrumentation, methods used, analyte recovery, reagents, cutoffs, and populations screened, which can make result comparisons problematic [10,24]. However, these differences are the reason harmonization is needed.

Our objective in this study was to harmonize enzyme activities for PD and MPS I analyzed using Digital Microfluidics (DMF) or Tandem Mass Spectrometry (MS/MS) across laboratories by two methods: using quality controls and regression equations, and MoM. Detailed descriptions of DMF and MS/MS lysosomal disorders (LD) screening have been described elsewhere [20,30,31,32,33]. We wanted to determine whether the same outcomes would be achieved: the correct identification of specimens exhibiting low or deficient activities (positive cases) and specimens exhibiting normal activities (negative cases). To our knowledge, this is the first study to harmonize GAA and IDUA enzyme activities from different platforms and across NBS laboratories. Results from this study demonstrate that harmonization has utility for interpreting enzyme activities to increase comparability in reported results across labs. Data generated from this study may also assist NBS laboratories in refining their enzyme activity cutoffs to achieve greater uniformity by reducing testing variabilities, decreasing false-positive rates, and consequently improving PPV.

## 2. Materials and Methods

Working with the Biochemical Mass Spectrometry Laboratory staff from the Newborn Screening Quality Assurance Program (NSQAP) at Centers for Disease Control and Prevention (CDC) and using data from the Newborn Screening Technical Assistance and Evaluation Program (NewSTEPs) [34], we contacted several laboratories screening for both MPS I and PD. Eleven states (California, Illinois, Indiana, Kansas, Maryland, Michigan, Minnesota, Missouri, New York, Ohio, and Virginia) agreed to participate in this study spearheaded by the Tennessee Department of Health, Division of Laboratory Services (herein referred to as TN). NSQAP provided quality control (QC) samples (*n* = 4) to serve as proof-of-concept calibrators with expected enzyme activities for GAA and IDUA at 0%, 5%, 50%, and 100% (herein referred to as A, B, C, and D QC pools, respectively) of that exhibited by pooled umbilical cord blood (Tennessee Blood Services, Memphis, TN, and LifeSouth, Gainesville, FL). We requested that each laboratory analyzed each QC pool six times as a set using their routine screening method. NSQAP also provided historical PT specimens (*n* = 6) exhibiting a range of activities: deficient (near zero), low (approximately 5%), or normal. For the purpose of this study, we defined deficient and low activities as positive for disease and normal as negative for disease. We asked each lab to analyze five replicates of each PT specimen. Manufacture of the QC pools and PT materials involved the use of leukocyte-depleted blood reconstituted with heat-inactivated, charcoal-stripped serum to achieve a hematocrit of 55% [35,36,37]. This mixture was subsequently restored either with cord blood from unaffected individuals in varying ratios to achieve the enzyme activities needed for the QC pools, or with lymphoblastoid cell lines derived from patients with PD or MPS I for the PT materials [35,36,37].

We asked the labs to run a blank (*n* = 14) after each set of QC pools and after each set of PT specimens to ensure that blank signals were below the A QC pool (the lowest calibrator). We required labs to report the signals from the blank wells, the enzyme activities from the QC pools, and the activities of the PT specimens in units of μmol/L/h for GAA and IDUA. We encouraged the labs screening for other LD, namely Fabry, Gaucher, Niemann-Pick, MPS II, and Krabbe, to report their respective enzyme activities, although the focus of this study was solely on PD and MPS I. Additionally, we asked labs to report the following information: enzyme cutoffs, daily median or mean for the day of analysis, average monthly median or mean enzyme activities for the 30-day period in which the specimens were analyzed, screening method, whether the test was multiplexed with non-LD analytes, incubation times and temperatures, and instrumentation. We converted the reported percent of the daily mean or daily median cutoffs into units of μmol/L/h for same-scale comparisons. To compare results, we grouped the labs based on instrumentation and reagents used.

We constructed box and whisker plots for GAA and IDUA using the reported raw enzyme activities from the PT specimens for each lab. As previously described [24], we took the natural logarithm (herein referred to as log) of the reported QC enzyme activity results (*n* = 24, 6 replicates of 4 QC pools) from all laboratories and constructed simple linear regression equations with TN as the reference laboratory. We compared the QC pools (initial and log harmonized) and subsequently evaluated each lab’s average blank signal for potential exclusion from regression equations based on whether this average was greater than the A QC Pool. Harmonization of PT values was performed using the regression parameters specific to each laboratory and enzyme, as previously described [24], and also by using MoM. We calculated the MoM by dividing the reported PT enzyme activities by the daily median and the monthly median for each laboratory. We did the same for the cutoffs. Lastly, we constructed box and whisker plots to display values from the log harmonization and the MoM. Each laboratory was deidentified except TN (Lab K), and data analyses were performed using statistical software R v 4.0.2 [38]. To determine whether the results for the PT specimens for each participating NBS lab were similar to the expected NSQAP result, we compared the raw enzyme activities, the log-harmonized enzyme activities, and MoM to each lab’s respective raw, log-harmonized, and MoM cutoffs.

## 3. Results

### 3.1. Characteristics of Participating Laboratories

Of the participating labs (*n* = 12), five used the FDA-cleared DMF Seeker® platform manufactured by Baebies, Inc., whereas seven used MS/MS. Of the MS/MS labs, five used a laboratory-developed assay with consumer-contracted reagents (CCR) and two used the FDA-cleared NeoLSD™ assay kit manufactured by Perkin Elmer Wallac, Inc. (PE, Turku, Finland). Two MS/MS labs used Waters Xevo TQD, (Milford, MA, USA), one used Waters Xevo TQ-MS, two used Waters Acquity TQD, and two used Q-Sight by PE. All MS/MS labs used flow injection analysis (FIA) except one that used ultra-high-performance liquid chromatography (UPLC) to separate enzyme substrates and products before analysis by MS/MS. All the labs using DMF and three labs using MS/MS routinely report patient screening results in units of μmol/L/h. Three labs using MS/MS (including TN) report results as a percent of the population daily median, whereas one lab reports results using percent of the population daily mean. One lab multiplexed their PD and MPS I screening with X-ALD.

Regarding incubation times for the labs using MS/MS, five labs reported overnight incubation (between 17 and 18 h) and two labs reported same-day incubation (3.5 h and 3.0 h). For the DMF labs, each performed an extraction of the DBS for 30 min at ambient temperature followed by sample injection into a cartridge with incubation for one hour at 37 °C. Regarding incubation temperatures, all MS/MS labs reported incubations at 37 °C.

### 3.2. GAA and IDUA Raw Enzyme Activities and Harmonization

The six PT specimens exhibited activities from deficient to normal as shown in Table 1. While we noted intra- and inter-laboratory variability for all PT specimens reported, we selected Blind-1, Blind-2, and Blind-3 to illustrate the performance of the assays (Figure 1). Labs using MS/MS reported overall lower enzyme values compared to the labs using DMF. In some cases, specimens with normal activities exhibited two-fold or greater values when analyzed by DMF as compared to MS/MS (Figure 1). For the two different MS/MS platforms (FIA-MS/MS and UPLC-MS/MS), and regardless of the source of reagents used (CCR or PE), the raw enzyme values were less variable than what was reported for DMF (Figure 1a–f). In addition to Blind specimens 1, 2, and 3, these observations held for the remaining PT specimens.

We observed that cutoffs varied for each lab (Figure 1g,h). Labs correctly identified activities for both GAA and IDUA as expected for all PT specimens except Blind-3. Six of seven MS/MS labs (B, C, G, I, J, and TN) reported enzyme activities for Blind-3 marginally above their respective cutoffs for IDUA with results classified as negative for disease. All DMF labs reported Blind-3 activity below their respective cutoffs, so results were classified as positive for disease (Figure 1).

Figure 2 displays scatter plots with linear regression lines for two laboratories (Lab A and Lab E) using Tennessee as the reference lab. The quantitative results for the QC pools from these three laboratories were different, as expected. Figure 2a,c show raw results for the four QC levels (i.e., A, B, C, and D), and it was apparent that the assumption of constant variance was not true. Therefore, the QC results from each lab and Tennessee were log-transformed to obtain constant variation across all QC pools (Figure 2b,d). This same workflow was applied to every laboratory’s QC results, and parameters from the log regressions were used in the harmonization of PT results as previously described [24]. The coefficient of determination (r2) for the log regressions either improved or remained unchanged for over 50% of labs for both GAA and IDUA (Figure 2 and Appendix A— Table A1). In regressions using both raw and log-transformed results, we observed outliers in the A, C, and D pools of some labs as observed in Figure 2.

We noted variability for the blank signal averages compared to the A QC pool averages across labs and platforms. Laboratories using MS/MS, had an average blank signal ≤ 0.02 μmol/L/h for GAA except for Lab A and TN (0.59 and 0.06 μmol/L/h, respectively). Nonetheless, all blank signal averages for the MS/MS labs were less than the average values of the A QC pool. The same was true for the IDUA blank replicates and these average values ranged from 0 to 0.23 μmol/L/h. For laboratories using DMF, the average GAA blank signals ranged from 0.01 to 1.57 μmol/L/h, whereas the average signal for the IDUA blanks ranged from 0.23 to 2.19 μmol/L/h (Appendix A—Table A2). We observed that the average blank signal for GAA exceeded the values reported for the A QC pool for Labs F and L, whereas for IDUA, values were exceeded for Lab F only. Since the majority of labs had blank signal averages less than the A QC pool, we included the A QC pool in our calculations and proceeded with harmonizing the log values of the PT samples.

As observed in Figure 3, GAA and IDUA enzyme activities were brought into closer agreement for most labs with intra- and inter-laboratory variability minimized overall, especially for the PT specimens that had expected low or deficient activities. We noted that for the log harmonized Blind-2 GAA PT specimen, Lab A exhibited notably higher enzyme activities compared to its respective raw enzyme activities. Additionally, log harmonization did not change the classification of results for Blind-3 for IDUA for any of the labs. As expected, the cutoffs became more similar after harmonization across the different platforms and methods.

### 3.3. Harmonization Using Daily Mom and Monthly Mom

Harmonization by MoM using the daily median enzyme activities reported by each laboratory minimized the intra-laboratory and inter-laboratory variability as enzyme activities were brought into closer agreement across platforms (Figure 4). This was also observed when we used the monthly values for MoM calculations (Appendix A—Figure A1). Neither the daily MoM nor the monthly MoM changed the result classification for Blind-3 for IDUA for any of the labs. Upon visual inspection, there were no observable differences when comparing the daily MoM and the monthly MoM for each lab (Appendix A—Figure A1).

## 4. Discussion

Due to increased emphasis on QI processes [1,2] and the recommendation for labs to adopt uniform cutoffs [3,4], we set out to harmonize enzyme activities for PD and MPS I from 12 NBS laboratories. It was also our intent to determine whether results could be correctly classified as positive or negative for these disorders, regardless of differences in methods or instrumentation. Contrived specimens were used for this study, since providing the same patient specimen to be analyzed in replicate by multiple labs was impossible. Raw enzyme activities for laboratories using the DMF platform exhibited greater variability in the replicates for GAA and IDUA compared to the activities for laboratories using the MS/MS platforms. MS/MS assesses enzyme activity by measuring the amount of product generated when an enzyme reacts with a synthetic substrate [33,39]. While DMF also measures enzyme activity, it does so by use of endpoint fluorescent substrates [40]. These differences in methods likely explain the variations in the observed enzyme activities.

Because each laboratory establishes its own cutoffs [10,12], the reported cutoffs varied (Figure 1). This variation is consistent with the cutoffs reported by NSQAP in its 2021 Annual Summary Report for domestic labs screening for IDUA and GAA [37]. For labs using DMF, cutoffs for GAA and for IDUA ranged from 6.60 to 10.0 μmol/L/h (median = 8.70; mean = 8.56) and 3.94 to 5.77 μmol/L/h (median = 4.90; mean = 4.79), respectively. For users of MS/MS, the GAA cutoffs ranged from 1.0 to 2.12 μmol/L/h (median = 1.97; mean = 1.88), whereas the IDUA cutoffs ranged from 0.57 to 1.80 μmol/L/h (median = 1.19; mean = 1.16) [37]. Since cutoffs are influenced by the varied populations which are being screened, the differences in the screening platforms [24], and goals to minimize detection of pseudodeficiencies and carriers, these considerations may explain the results we observed for Blind-3. As noted, we classified Blind-3 raw and harmonized results for six of seven labs using MS/MS as negative for IDUA, while the results for all DMF labs were classified as positive. It is not surprising that harmonization did not change how results were classified for any of the contrived specimens. Harmonization scales data point to different enzyme values based on a laboratory’s QC results, and since enzyme activities for the PT specimens and cutoffs undergo the same harmonization, their relative positions did not change. In other words, if a particular specimen exhibited an enzyme activity either above or below the cutoff, after log harmonization or harmonization by MoM, that enzyme’s activity would still be either above or below the cutoff. Since harmonization has been shown to minimize inter- and intra-laboratory variability [17,24,27,28,29], it may be best to use harmonized data when setting cutoffs. It would be interesting to see if cutoffs set in this manner are more consistent across laboratories and if cases of discordant PT results, such as those observed for Blind-3, are minimized. It would also be interesting to observe the effects of using harmonized data when performing CLIR analysis to determine whether a screening result is deemed positive or negative.

Blind-3 was included in this study, although it was a PT specimen for Krabbe disease that had been designed to demonstrate deficient activity for galactocerebrosidase (GALC) but normal activities for other enzymes (i.e., GAA and IDUA). Blind-3 was prepared using a lymphoblastoid cell line derived from a true Krabbe specimen [35,36,37]. Cell lines are intended to have only one deficient enzyme, but in this case the cell line used had low IDUA levels, giving rise to positive reports from some labs and not others. NSQAP is currently investigating producing PT specimens by addition of recombinant enzymes, therefore bringing activities into a higher range of normal to distinguish them from PT target enzyme and prevent problems of concomitant positive results.

Regarding multiplexing with a non-LD analyte, we could make no comparisons across labs as only one lab reported multiplexing under this circumstance (C26 for X-ALD). There also seemed to be no marked differences observed between the use of FDA-cleared reagents (PE) and the CCR for the MS/MS laboratories. Five MS/MS laboratories reported incubation times between 17 and 18 h. Two MS/MS labs reported incubation times of 3 and 3.5 h; however, these labs only analyzed for PD and MPS I. Shorter incubation times for MS/MS have been documented as effective in separating normal enzyme activity from low enzyme activity for PD and for MPS I only [41], as was the case in this study; however, it is reported to be less effective for Krabbe disease due to the slower reacting GALC enzyme [30]. Hence, longer incubation times are required for proper detection of Krabbe disease [30] and to minimize false positives. DMF has a short assay time starting with a 30-minute extraction phase and then incubation of the analytical cartridge [40]. The three-hour analysis time for each cartridge after sample preparation [20] gives DMF an advantage over the MS/MS platform, which requires further processing and analysis after incubation is complete. Shorter assay times equate to faster results and faster reporting, thereby facilitating a quicker diagnosis and subsequent intervention for affected infants [20,30,41].

Neither simple linear regression nor regression using log values removed the observed variability in the QC pools. In fact, using the log values revealed outliers which were more apparent for the A QC pool and, for some labs, the B QC pool. This situation can result from the differential effect of taking the logarithm of small numbers versus large numbers, or these outliers may be indicative of imprecision [30,42]. This imprecision could be attributed to differences in instrumentation, reagents used, and methodologies—all of which have been cited as barriers to harmonization [10,24].

There was variability in how the MS/MS labs analyzed their blanks. Some used blank filter paper punches combined with the internal standard/substrate cocktail, while others used the cocktail alone. This difference may have contributed to the variable activities observed for these blank replicates. For routine testing, some MS/MS labs run up to four blank wells and only monitor these wells for activity, whereas others subtract the average value of the blanks from the activities obtained for patient specimens prior to reporting. In this study, we did not subtract the blank signal averages, but instead used these to gauge acceptability of the lower activity QC pool. Lab A and TN had higher blank signal averages compared to those of the other MS/MS labs and this may be attributed to non-enzymatic breakdown of substrate with the formation of product (in-source fragmentation) as is common with electrospray ionization in the screening for LDs [23,35,43].

During the extraction process for the DMF platform, two wells are routinely designated as blank wells containing only extraction buffer (i.e., without filter paper) which are subsequently loaded into certain corresponding positions on the analytical cartridge [40]. In addition to these manufacturer-designated blank wells, labs using DMF included 14 locations on their extraction plates and cartridges as blanks for the purpose of this study. There was variation as to how these additional blanks were treated. Two labs punched blank filter paper into their 96-well extraction plate; one lab only added extraction buffer; and two labs punched the DMF kit QC base pool (QCBP) filter paper into these wells as recommended by the manufacturer for otherwise “empty” wells [40]. After analysis of the cartridge, data from the manufacturer’s designated blanks are automatically subtracted from all specimen wells by the DMF software prior to reporting the specimen enzyme activities [40]. Additionally, a hemoglobin correction factor is applied to all sample and QC wells based on the expectation that hemoglobin quenches fluorescence [44,45]. Our comparison of the average of the 14 blank replicates to the A QC pool did not include a comparison to the manufacturer’s two designated blanks as we did not have this information. However, with the automated subtraction of these designated blank values and the application of the hemoglobin correction factor, the average signals exhibited by the 14 blank replicates for two of five DMF labs approximated or even exceeded the signals obtained for the A QC pool (the lowest calibrator). Intrinsic fluorescence of the substrates and the conversion of a small amount of substrate to product may have also contributed to the high blank averages that we observed [43,44,46]. We assumed that if QCBP had not been used, the 14 blank replicates would have been similar across DMF labs, hence the reason for the retention of their A QC pools in the regression equations.

While the log harmonization brought enzyme activities for the PT specimens into closer agreement, this methodology did not achieve uniformity of results for specimens with normal enzyme activities. Since low enzyme activity gives indication of possible disease, we could argue that harmonization is unnecessary for specimens with normal enzyme activities; however, it begs the question of what enzyme activity level would negate performing harmonization of the data. This consideration leads us to a limitation of this study: the use of QC samples as proof-of-concept calibrators. We had four QC pools with expected enzyme activities at 0%, 5%, 50%, and 100% of that exhibited by pooled blood. Since QCs are produced using pooled blood, while PT specimens are made using cell lines from patients with disease, “normal” enzyme activity levels (i.e., no deficiency) in a PT specimen can exceed that of the 100% QC pool. Consequently, extrapolation beyond the highest QC pool (100% activity) contributed to inaccuracies, making the harmonization unreliable for some PT specimens. We suggest that future studies include actual calibrators, but what is unknown is the upper limit of the highest-level calibrator. We also suggest additional calibrators ranging from 0 to 30% of pooled blood activities, since most participating labs reported enzyme activity cutoffs falling within this range. These low calibrators would be ideal since detection of PD and MPS I is contingent on recognizing low enzyme activity and classifying it as such. Having additional calibrators may reduce outliers and facilitate a better fit for the regression line at the low and high ends, minimize extrapolation, and perhaps yield improved results from what was achieved for this study. NSQAP is currently investigating the manufacturing of dried-blood-spot-based quality assurance materials using solely recombinant enzymes (i.e., eliminating the need for cell lines or umbilical cord blood products) that may be more suitable when used for harmonization studies.

On visual inspection of the MoM plots, we observed no marked differences for any lab when comparing their daily median activity versus their monthly median activity (Appendix A—Figure A1), and the lack of variability likely indicates that the methodologies used are robust. Since laboratories also have redundancy with their instrumentation, it is likely that the reported monthly median activity covers data from more than one instrument, also giving indication of instrument stability over time. We could not make a comparison between the daily and monthly MoM for Lab E since they had just begun routine screening for PD and MPS I at the outset of this study and had insufficient data to calculate and report a monthly MoM. Future studies should include these data and perhaps expand to include a median beyond one month.

Another limitation is that our sample size was small in comparison to the number of NBS laboratories currently screening for PD and MPS I in the US. As of February 2022, 33 states analyze for PD, while 31 analyze for MPS I [34]. Future studies may also be needed as more laboratories begin screening for these disorders or change platforms. Additionally, as more LDs are added to the RUSP, these too may benefit from harmonization. GAA and IDUA enzyme values were harmonized using TN as the reference. TN uses FIA-MS/MS combined with PE NeoLSD™ reagents for screening. It could be that harmonizing with a CCR FIA-MS/MS laboratory, a DMF laboratory, or a UPLC-MS/MS laboratory would yield different results than what were observed for this study. We harmonized data using values expressed as μmol/L/h; using % of the daily mean or median activity is another way to harmonize results and should be studied. Another harmonization study may be warranted as more suitable materials become available for use as calibrators. Using NSQAP as the reference as opposed to using a participating laboratory should be explored.

## 5. Conclusions

Both DMF and MS/MS platforms have merit for NBS of PD and MPS I; however, varied enzyme values are obtained by these two methods and by the same method performed by different laboratories. When harmonization is performed by log regression or MoM, these apparently disparate values come into reasonable agreement. However, despite this agreement, harmonization does not change how a value would be reported (positive or negative) as this is dependent on cutoff placement, as observed with Blind-3. Harmonization and MoM still have practical value in comparing assay performance across different laboratories. However, to avoid the inaccuracies associated with extrapolation, we recommend that calibrators extend above and below the values used here and that these calibrators be supplied by one common source. These calibrators could also be used during new assay validations or verifications so that harmonized values are the basis for establishing cutoffs. However, even if all labs were harmonized in data they produce, the difficulty still remains in deciding cutoff placement, as this may be greatly affected by case definitions and the overall goals of the screening program. Routine use of calibrators could also facilitate periodic comparisons of enzyme activities across labs and across platforms, thus providing greater uniformity to NBS for LD.

## Figures and Tables

**Figure 1 IJNS-09-00011-f001:**
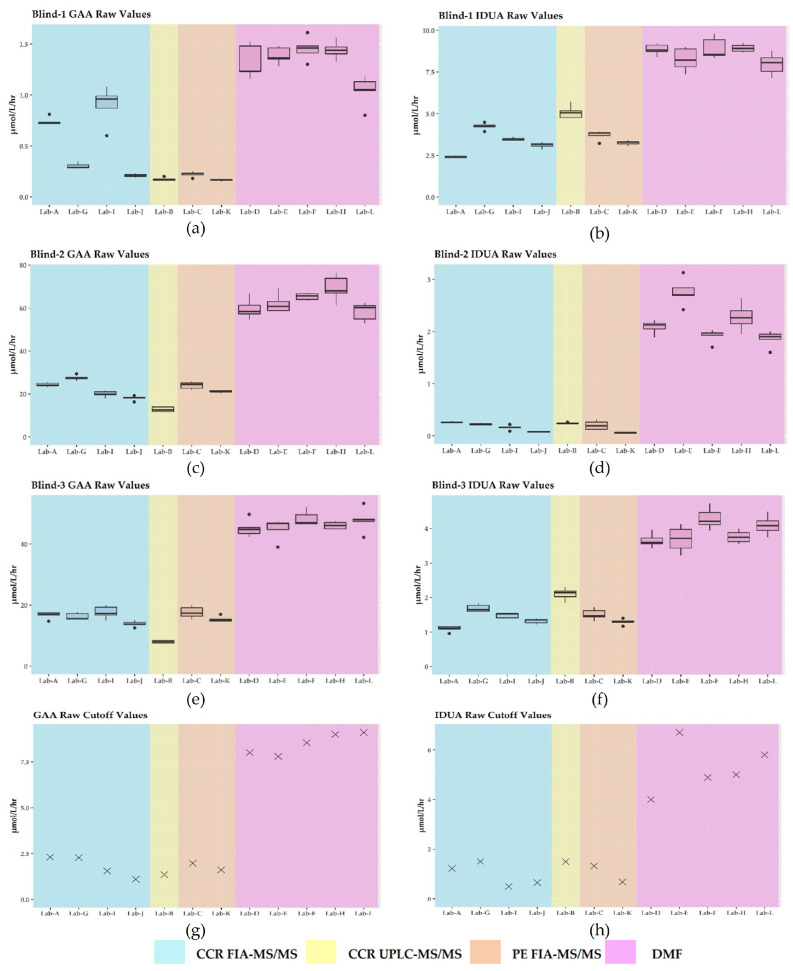
Plots for the raw enzyme activities by laboratory: (**a**) deficient GAA specimen; (**b**) normal IDUA specimen; (**c**) normal GAA specimen; (**d**) deficient IDUA specimen; (**e**) normal GAA specimen; (**f**) low IDUA specimen; (**g**) GAA cutoff; (**h**) IDUA cutoff.

**Figure 2 IJNS-09-00011-f002:**
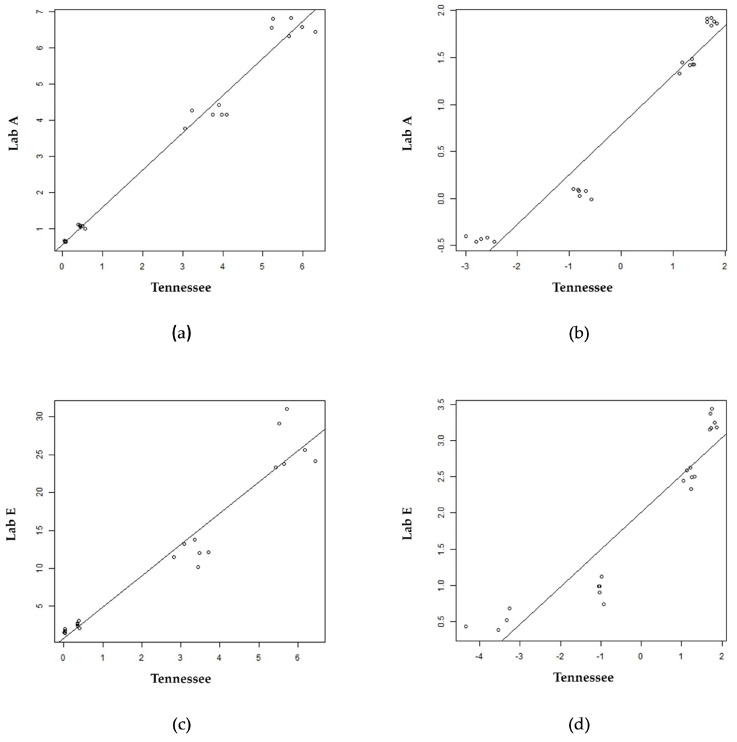
Scatter plot overlayed with linear regression line of raw enzyme and log enzyme values for the QC pools (A, B, C, and D pools moving up the line) for two labs using Tennessee as a reference lab: (**a**) GAA raw; (**b**) GAA log; (**c**) IDUA raw; (**d**) IDUA log.

**Figure 3 IJNS-09-00011-f003:**
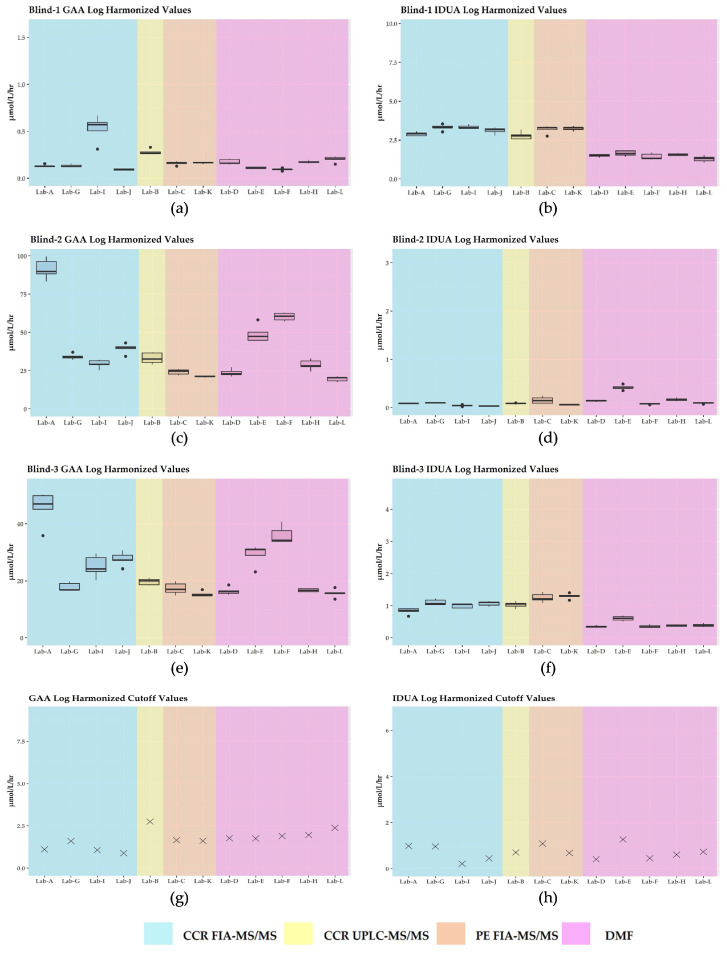
Plots for log harmonization of enzyme activities by laboratory: (**a**) deficient GAA specimen; (**b**) normal IDUA specimen; (**c**) normal GAA specimen; (**d**) deficient IDUA specimen; (**e**) normal GAA specimen; (**f**) low IDUA specimen; (**g**) GAA cutoff; (**h**) IDUA cutoff.

**Figure 4 IJNS-09-00011-f004:**
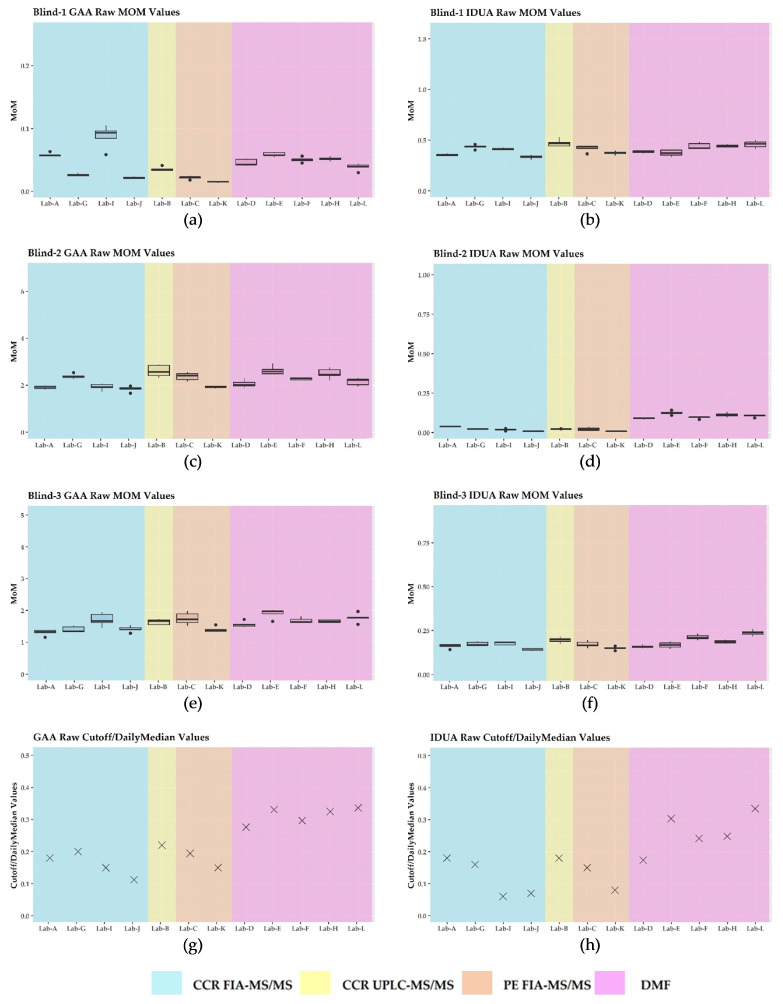
Plots for daily MoM by laboratory: (**a**) deficient GAA enzyme activity specimen; (**b**) normal IDUA enzyme activity specimen; (**c**) normal GAA enzyme activity specimen; (**d**) deficient IDUA enzyme activity specimen; (**e**) normal GAA enzyme activity specimen; (**f**) low IDUA enzyme activity specimen; (**g**) GAA MoM cutoff; (**h**) IDUA MoM cutoff.

**Table 1 IJNS-09-00011-t001:** Expected enzyme activities for PT specimens for GAA and IDUA.

PT Identifier	^1^ GAA (PD) Activity	^1^ IDUA (MPS I) Activity
Blind-1	Deficient	Normal
Blind-2	Normal	Deficient
Blind-3	Normal	Low
Blind-4	Deficient	Deficient
Blind-5	Low	Low
Blind-6	Normal	Normal

^1^ Deficient or low activities are considered positive for disease while normal is negative for disease.

## Data Availability

Deidentified raw participant data are available on request from the corresponding author.

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
