# Peer review of "Harmonization of Newborn Screening Results for Pompe Disease and Mucopolysaccharidosis Type I"

_2409-515X, 2023, doi:10.3390/ijns9010011_

Round 1
Reviewer 1 Report
The paper describes harmonization of NBS results for Pompe disease and MPS1 across State NBS labs that use either MS/MS or DMF methodologies. One of the labs was used as reference and regression analysis and MoMs were used to harmonize enzyme data analyzed. Variations in cut-off were detected between the two platforms used in the testing methodologies (DMF versus MS/MS) and despite log-transformation an use of MoMs. Authors described succinctly some of the limitations of the study and future remedies including sample size considerations to help with harmonization testing across all platforms used for NBS.
The paper is well-written and the methodologies were sound. As many laboratories start to offer testing for Pompe disease, MPS1 and other diseases to be added, a follow-up study on harmonization across several of these laboratories will be appropriate to improve sample size and hopefully help improve cut-points and bring uniformity to NBS testing. Also using a non-participating laboratory as reference will be desirable as the authors correctly alluded to.
Author Response
Hello Reviewer 1,
We appreciate the time you took to review this manuscript. You had no questions or required no changes so we cannot say anything but thank you for your honest assessment.
Christine
Reviewer 2 Report
I consider the publication of this article to be interesting as harmonization studies are necessary to improve inaccuracies even if they do not improve the qualitative interpretation of positive or negative samples. In addition, the study provides information on the advantages or disadvantages of using different methods or protocols, which may help future laboratories that have not yet included these diseases in their panels.
I wanted to consider that the harmonization by MoMs is good because it includes the daily variability and does not vary with the monthly variability of the laboratories, thus demonstrating the robustness of the methods. But to be able to compare between laboratories, for example in the external quality control of the ERNDIM program for lysosomal enzyme studies a previous control is analyzed and the data of the following samples are given as a result of the % of the activity with respect to the first control and depending on % is given as a positive or negative sample for a particular enzyme. This % provide by the labs in the ERNDIM is more comparable than the raw data in umol/l/hour between laboratories, as some show higher values than others, as it happens with the MS/MS and DFM methods reported in this article. I assume and also is mentioned that the laboratories to report their cut-off and results are expressed in umol/l/h, but it not seems true, as you commented in the article that “you converted the reported percent of the daily mean or daily median cutoffs into units of µmol/L/h for same scale comparisons. I don’t know if it would be better to compare these % respect daily mean or median directly” (page 3, line 107) .Could discuss it? Maybe could the qualitative interpretation of Blind 3 change? Could you mention what advantage or disadvantage it would have to compare with this % or the method used in the article with MoMs in umol/l/h?
Author Response
Hello Reviewer 2,
I speak for myself and the remaining authors when I say we appreciate the time you took to review this manuscript. To address your points, our response is as follows:
- We acknowledge that % of activity be it daily median or daily mean could be another way to harmonize data, however it was not evaluated in this study.
- We have not tried this type of harmonization and therefore cannot hypothesize on the advantages/disadvantages of that approach.
- We used µmol/L/h to perform same scale comparisons as you pointed out because eight labs out of 12 routinely report using µmol/L/h and have cutoffs expresed as µmol/L/h (See lines 137-139).
- We asked that labs report values from the QC and PT specimens in µmol/L/h and to give us their cutoff (whether it be in µmol/L/h or % of the daily median, or % of the daily mean). Obtaining the daily medians/mean facilitated the conversion to µmol/L/h for the cutoff for those four labs.
- Lastly we have added into the manuscript (See lines 351-353) that another way to harmonize could be use of the % daily mean or daily median and should be studied.
Best Regards,
Christine